# A Real-Time Fault Early Warning Method for a High-Speed EMU Axle Box Bearing

**DOI:** 10.3390/s20030823

**Published:** 2020-02-04

**Authors:** Lei Liu, Dongli Song, Zilin Geng, Zejun Zheng

**Affiliations:** State Key Laboratory of Traction Power, Southwest Jiaotong University, Chengdu 610031, China; liul@my.swjtu.edu.cn (L.L.); gengzilin@my.swjtu.edu.cn (Z.G.); zhengjuner@my.swjtu.edu.cn (Z.Z.)

**Keywords:** high-speed EMU, axle box bearing, real-time, early warning, LSTM, iForest

## Abstract

An axle box bearing is one of the most important components of high-speed EMUs (electric multiple units), which runs at a very fast speed, suffers a heavy load, and operates under various complex working conditions. Once a bearing fault occurs, it not only has an enormous impact on the railway system, but also poses a threat to personal safety. Therefore, there is significant value in studying a real-time fault early warning of a high-speed EMU axle box bearing. However, to our best knowledge, there are three obvious defects in the existing fault early warning methods used for high-speed EMU axle box bearings: (1) these methods based on vibration are extremely mature, but there are no vibration sensors installed in high-speed EMU axle box because it will greatly increase the manufacturing cost; (2) a TADS (trackside acoustic device system) can effectively detect early failures, but only a portion of railways are equipped with such a facility; and (3) an EMU-ODS (electric multiple unit onboard detection system) has reported numerous untimely warnings, along with warnings of frequent occurrence being missed. Whereupon, a method is proposed to realize the fault early warning of an axle box bearing without installing a vibration sensor on the high-speed EMU in service, namely a MLSTM-iForest (multilayer long short-term memory–isolation forest). First, the time-series data of the temperature-related variables of the axle box bearing is used as the input of MLSTM to predict the axle box bearing temperature in the future. Then, the deviation index of the predicted axle box bearing temperature is calculated. Finally, the deviation index is input into an iForest algorithm for unsupervised classification to realize the fault early warning of an axle box bearing. Experimental results on high-speed EMU operation data sets demonstrated the availability and feasibility of the presented method toward achieving early fault warnings of a high-speed EMU axle box bearing.

## 1. Introduction

A high-speed EMU (electrical multiple unit) axle box bearing is one of the most crucial but vulnerable components, which rotates with a very high-speed, suffers a heavy load, and operates under various complex work conditions. Once a bearing failure occurs, it not only directly threatens the safety of passengers, but also imposes a great impact to the railway dispatching system. 

According to statistics, in China, the number of high-speed EMUs with a design speed over 350 km/h is 1441, the number of high-speed EMUs in service is 3114, and the number of axle box bearings of different train types ranges from 64 to 136. In 2018, more than 80 wheelsets were replaced due to bearing failure. Figure 1a displays a CRH380B high-speed EMU that is under maintenance. Figure 1b shows that a wheelset is removed from the high-speed EMU, and the axle box bearing is inside the axle box. A broken and dissembled axle box bearing is shown in Figure 1c.

In order to warrant the safe operation of high-speed EMUs, it is crucial to predict the healthy operation of an axle box bearing to prevent any catastrophic accident caused by axle box bearing failure. 

A fault early warning is a part of condition monitoring and different methods are used for the diagnosis of rotary machines, such as bearing, gears, and motors. According to data sources, they can be broadly classified as vibration-based data [1,2,3,4,5,6], acoustic-based data [7,8], temperature-based data [9,10], and fusion-based data [11,12,13,14,15,16,17]. 

Among these, the vibration-based method is the dominant technique used to inspect a potential defect of a rotary machine. With the continuous rotation of rolling elements, the load zone is constantly changing, which will generate normal vibration. However, the existence of a defect constitutes a series of periodic impacts, which lead to an enormous increase in the vibration level. The vibration-based method can be divided into one of the following categories: time-domain analysis techniques [18], frequency-domain analysis techniques [19], and time–frequency analysis techniques [20]. The vibration-based technique is quite mature, but for high-speed EMUs in China, vibration sensors have not been installed in axle box bearings because assembling the acceleration sensors for each axle box bearing will greatly increase the manufacturing cost of each high-speed EMU.

A TADS (trackside acoustic device system) is an acoustic-based system used to implement a bearing early warning system by giving an early indication of a possible internal bearing fault. Anderson and McWilliams [21] describe the development of a TADS and conclude that a TADS not only has a good performance regarding capturing different types of defects, but also has the capability to identify defects before severe faults occur. Montalvo et al. [22] have found that a TADS misses a few catastrophic defects, leading to serious derailments; furthermore, only some of the lines in China have installed a TADS. 

The temperature-based method is based on the fact that when the mechanical system is in operation, it will inevitably generate heat due to the existence of friction, and different damage levels of the mechanical system will change the friction state, such that the mechanical system will display different thermal behaviors. Therefore, the temperature is a significant index that is used to monitor the state of a rotary machine. For instance, Harvey et al. [23] presented a condition monitoring method for rolling element bearings based on electrostatic wear monitoring and validated the presented method by monitoring the temperature rise. 

The temperature sensor is mounted on the axle box and once the temperature reaches the threshold, the EMU-ODS (electric multiple unit onboard detection system) assembled on the train will give an alarm. The warning of EMU-ODS is untimely, and even worse, some warnings are frequently missed. The defects of the EMU-ODS are obvious, and untimely warnings will lead to the lack of an emergency response time, and a missed warning is a potential security risk that will increase the workload of manual inspection.

In consideration of manufacturing cost, the acceleration sensor is not installed in the axle box, while both TADS and EMU-ODS have various deficiencies. However, before leaving the factory, a high-speed EMU is equipped with many other sensors, such as sensors to monitor the axle box bearing temperature, sensors to record the traction power of motors, sensors to report the train speed, sensors to detect the ambient temperature, etc. The bearing temperature has a direct correlation with these factors. The change of each factor is closely related to the state of the bearing, and is finally reflected by the change of the bearing temperature. In order to improve the reliability of real-time fault early warning methods, investigating the full historical route of sequential data may be a promising research direction. 

A RNN (recursive neural network) and an LSTM (long short-term memory) [24] are deep learning algorithms based on the machine learning algorithm, which have been shown to be very effective for solving sequence problems [25]. It has also been shown that LSTM performs better than RNN for long-term dependency problems. Inspired by this, this study explored a high-speed EMU axle box bearing fault early warning system by employing machine learning. 

The fault early warning based on machine learning is fundamentally a classification problem, which can be divided into supervised and unsupervised learning algorithms. If a supervised machine learning algorithm is adopted, the work to label is needed, which involves a large amount of manual work and there will also be a subjective factor that affects the labeling result. Since the iForest (isolation forest) algorithm [26] is proved to be an effective unsupervised classification method, the MLSTM-iForest algorithm was proposed. MLSTM is the abbreviation of multilayer long short-term memory, which is mainly used for processing time-series data to predict the temperature of the axle box bearing, and iForest is primarily used for classification. The main contributions of this paper can be summarized as follows:(a)The fault early warning of a high-speed EMU axle box bearing could be implemented by using the existing data collected by sensors assembled on the train without installing acceleration sensors, which could save a lot of money.(b)Compared with the EMU-ODS, the MLSTM-iForest greatly reduced the false alarm rate, which immensely diminishes the workload of manual verification.(c)The existing condition monitoring method based on LSTM is a supervised algorithm, whereas MLSTM-iForest is an unsupervised deep learning algorithm that can eliminate a huge workload and the subjective influence of labeling.(d)In order to eliminate the instantaneous abnormal data collected in the operation that caused the false warning, a degree of deviation index was proposed.

The remainder of this paper is organized as follows. Section 2 introduces the background of the presented model. The axle box bearing real-time fault early warning using MLSTM-iForest is presented in Section 3. A series of comparisons and the associated discussion is given in Section 4. In the end, Section 5 gives the conclusions.

## 2. Model Background

### 2.1. LSTM Model

Time-series data refers to the data arranged in chronological order and one obvious characteristic of this kind of data is that there is a strong correlation between the data before and after. RNN is a kind of deep learning network that is particularly suitable for dealing with time-series problems. RNN includes input layer, hidden layer, and output layer. It controls the output through an activation function and links layers to layers by weights that were obtained while the model learned through training.

LSTM, a variant of RNN, was proposed by Hochreiter et al. [24] in 1997. RNN can only have a short-term memory because of the vanishing gradient. An LSTM network solves the vanishing gradient problems to a certain extent through delicate gate control, which can effectively learn information about long-term dependencies. LSTM has achieved marvelous results when dealing with a large number of time-series problems and has attracted increasingly more attention from researchers. 

Figure 2 shows the structure of the LSTM model. The following meanings of the symbols are demonstrated in the figure: subscript *t* stands for “at time *t**,*” *x* is the input matrix, *h* is the hidden state, *c* is the cell state, *f* stands for the forgetting gate, *y* is the output matrix, *i* is the input gate, *o* is the output gate, *c`* is the candidate memory unit, *σ* is the sigmoid activation function, and *tanh* is the hyperbolic tangent activation function. “*” is a multiplication operation and “+” is an addition operation.

One of the cores of the LSTM model is the cell state, which stores long-term memory and transmits an input forward, thereby solving the long-term dependencies problems. Another core of the LSTM model is the gate structure with a different function to update the cell state. The so-called gate consists of a sigmoid activation function and a multiplication operation. The output of the sigmoid function is a continuous function from 0 to 1. If the output of the sigmoid function is 0, it is equivalent to closing the gate completely and not allowing any data to pass through. If the output of the sigmoid function is 1, this corresponds to opening the gate completely, permitting all data to go through.

There are three main stages in an LSTM. The first one is the forgetting stage, which determines what information in the cell state needs to be lost. Specifically, the hidden state of the previous moment is spliced with the input of this moment, and then the spliced matrix is put into the sigmoid activation function, where the computational process of forgetting gate $f_{t}$ is shown in Equation (1). In the formula, “[*a,b*]” stands for *a* and *b* spliced together, *W* is the weight matrix, and *b* is the bias matrix. When the output of the forgetting gate is 0, the cell state of the previous moment is abandoned, and when the output of forgetting gate is 1, the cell state of the previous moment is preserved.
(1)$$f_{t}=\sigma (W_{f}*[h_{t-1},x_{t}]+b_{f})$$

The second stage is the memory stage, which consists of two parts. First, the input gate determines what values are to be updated. The input gate $i_{t}$ is shown in Equation (2). Second, a candidate memory unit matrix $c_{t}^{`}$ is created using a hyperbolic tangent activation function, which is shown in Equation (3). In the next step, it combines $i_{t}$ and $c_{t}^{`}$ to update the cell state, and the updated cell state $c_{t}$ is shown in Equation (4).
(2)$$i_{t}=\sigma (W_{i}*[h_{t-1},x_{t}]+b_{i})$$
(3)$$c_{t}^{`}=tanh(W_{c}*[h_{t-1},x_{t}]+b_{c})$$
(4)$$c_{t}=f_{t}*c_{t-1}+i_{t}*c_{t}^{`}$$

The third stage is the output stage. During this phase, the content of the output $y_{t}$ and the hidden state $h_{t}$ are based on the cell state $c_{t}$. The output gate is used to regulate which information is needed in the cell state, and the output gate $o_{t}$ is shown in Equation (5). The hidden state $h_{t}$ in Equation (6) can be obtained by multiplying $o_{t}$ and the hyperbolic tangent activation function on $c_{t}$. If it is required to output the results, $h_{t}$ is passed through another activation function. The sigmoid function is selected as the activation function to obtain the output $y_{t}$ and can be seen in Equation (7).
(5)$$o_{t}=\sigma [W_{o}*[h_{t-1},x_{t}]+b_{o}]$$
(6)$$h_{t}=o_{t}*tanh(c_{t})$$
(7)$$y_{t}=\sigma (W_{h}*h_{t}+b_{h})$$

### 2.2. Isolation Forest Algorithm

iForest is the abbreviation of isolation forest. It is an unsupervised abnormal detection method based on ensemble-learning. iForest is constituted by plenty of iTrees and the implementation procedure of an iTree is as follows:(a)Randomly select *ψ* samples from the dataset as a sub-sample and put them into the root node of the tree.(b)Randomly specify an attribute *q* from the current node and randomly generate a split point *p* between the maximum and minimum value of the attribute value.(c)A hyperplane is generated from the split point, and then the data space of the node is divided into two subspaces. Put less than *p* in the left subspace and put more than *p* in the right subspace.(d)Repeat steps (b) and (c) until the subspace cannot split or the node reaches the specified height.

The implementation of the procedure of iTree is repeated until *t* iTrees are created and the iForest algorithm is built. One-dimensional data is used to demonstrate the principle of iForest and it is shown in Figure 3. The dataset *D* = [*a, b, c, d, e, f*]*^T^* is assumed and a randomly selected sub-sample *D_sub_* = [*a, b, c, d, e*]*^T^* is put into the root node. Among them, the red dot represents abnormal data, the value of *a* to *e* is from small to large, i.e., *a* is the minimum and *e* is the maximum. First, a value *p* between *d* and *e* is randomly chosen, as shown in ${\mathrm{iTree}}_{1}$; then, the root node is divided into two sub-nodes, the left child root *C_l_* = [*a, b, c, d*]*^T^* and the right child root *C_r_* = [*e*]*^T^*. The left child root continues to split as above until it can no longer be segmented or it reaches the height limit. The data set *C_r_* is isolated very early because of its inseparability, that is, its path length to the root node is very short. Since there is a long distance for the interval between *d* and *e*, *p* has a very high probability to fall in this gap. The key to this algorithm is ensemble learning. Therefore, most *e*’s from *t* iTrees will be segmented very early. In other words, the average path of *e* in *t* iTrees will be very small, and the outlier is the point with a very small average path.

During classification, the sample data *x* traverses all *t* iTrees, and then calculates the height of *x* in each individual iTree. *E*(*h*(*x*)) is the average path length over *t* iTrees, which is obtained by calculating the average value of *x* at the height of each iTree. After obtaining the average path length of each test data, the abnormal score is calculated using Equation (8).
(8)$$Score(x)=2^{-\frac{E(h(x))}{c(n)}}$$
(9)$$c(n)=2H(n-1)-\frac{2(n-1)}{n}$$

In Equation (8), *h*(*x*) is the path length, *E*(*h*(*x*)) presents the average path length of *x* in iForest, *c*(*n*) is the average of *h*(*x*) given *n* to normalize *h*(*x*), *c*(*n*) is defined in Equation (9), and *H*(*n*) can be calculated using *ln*(*n*) plus Euler’s constant. Through Equation (8), *Score*(*x*) tends to 0.5 when *E*(*h*(*x*)) tends to *c*(*n*), *Score*(*x*) tends to 1 when *E*(*h*(*x*)) tends to 0, and *Score*(*x*) tends to 0 when *E*(*h*(*x*)) tends to *n* − 1. Therefore, a threshold can be set, and it is considered abnormal when the score is higher than the threshold.

iForest can solve swamping and masking effects well because of subsampling. The algorithm is neither based on distance nor density measurements; therefore, it can greatly reduce the memory consumption and is more suitable for engineering applications.

## 3. MLSTM-iForest Algorithm

### 3.1. Introduction of Experimental Data

The data of this paper comes from the real-time WTDS (wireless transmit device system) data collected by CRH380B. The function of a WTDS is to transfer the data gathered by dozens of different types of sensors installed on a high-speed EMU to the ground data center without delay. Because WTDS data is rich in a variety of real-time operation data from a high-speed EMU, it is indispensable to excavate and analyze WTDS data intensively. Through correlation analysis, it was found that four variables had a strong correlation with the axle box bearing temperature, which were train speed, motor traction power, ambient temperature, and train mass.

In Figure 4, (a) shows the axle box bearing temperature, (b) shows the ambient temperature, (c) shows the train mass, (d) shows the motor traction power, and (e) shows the train speed. All the data were collected in real-time by the WTDS. It can also be seen from Figure 4 that these four variables had a direct correlation with the axle box bearing temperature. The correlation coefficient and heat map are shown in Figure 5, which reveals that the axle box bearing temperature had a strong positive correlation with ambient temperature, carriage mass, motor traction power, and train speed. Therefore, these five parameters were taken as the input of the presented algorithm.

### 3.2. The Comparison between Different MLSTM Structures

Temperature is a direct indicator of the bearing condition. Once the bearing temperature is abnormal, it often indicates that the bearing has an extremely serious fault, and the bearing condition will deteriorate in a short time, which directly threatens the operational safety. Nevertheless, it is also of significant value if the fault early warning can be given as early as possible before the deterioration becomes serious. Therefore, this research aimed to establish a fault early warning system for an axle box bearing by predicting the temperature of the axle box bearing to reserve time for an emergency response. 

Although the LSTM network is very suitable for prediction, the prediction performance of a single-layer LSTM is not very good. Thus, a special multilayer LSTM structure was designed, named an MLSTM network, to reduce the prediction error. In order to improve the reliability and accuracy of the prediction, the input of the MLSTM should be variables that have a high correlation with the axle box bearing temperature. The change of any variable will directly reflect the change of the axle box bearing temperature. 

When an EMU-ODS alarm is raised, there is generally not enough time for an emergency response. Therefore, the MLSTM model was proposed based on deep learning to predict the future bearing temperature with the existing data of a period of time in history, and then use the unsupervised classification algorithm iForest to realize the axle box bearing fault early warning.

The MLSTM network structure takes the axle box bearing temperature, carriage mass, ambient temperature, motor traction power, and train speed from the past 40 min as the input of the model to predict the axle box bearing temperature for the next 6 min. The reason why the data of the past 40 min were selected as the model input to predict the temperature of the next 6 min is that after a large number of experiments, it was found that if the input time is too short, this will lead to overfitting and an unacceptable model prediction error. If the input time is too long, this will lead to overfitting and a poor generalization ability of the model. Although the shorter the prediction time is, the smaller the error is, this study aimed to extend the prediction time as much as possible when the error was acceptable. Since the temperature of the axle box bearing collected was accurate to 1 °C, after the experiment, 6-min MLSTM prediction results could effectively control the error within 1 °C, that is, the prediction error did not exceed the error of the collected temperature. 

First, the data from the training set was used to train the MLSTM model, and then the test set was used to verify the generation ability of the MLSTM model. In order to prevent the performance degradation caused by the imbalance of data set categories, the ratio of our training set to the test set was 1:1; the sample information of the training set and test set is shown in Table 1. Figure 6(a1) demonstrates the prediction performance of the one-layer LSTM using the test set, in which the blue line is the actual temperature of the bearing, while the orange line is the predicted temperature, and the right picture (Figure 6(a2)) is the local enlarged drawing. Since the collected temperature was accurate to 1 °C and the predicted temperature was accurate to two decimal places, there was a fluctuation in the predicted temperature. It can be seen from Figure 6(a1,a2) that the error between the actual temperature and predicted temperature was obvious. Figure 6(b1,b2) are the comparison curves of the predicted temperature curve and the actual temperature curve of the two-layer LSTM in the test set, from which the error showed a greater improvement than that from the one-layer LSTM. 

Figure 7(a1,a2) show the three-layer LSTM predicted axle box bearing temperature and actual axle box bearing temperature comparison curves in the test set, where the error was further improved, but the fitting degree was still insufficient, which was due to underfitting. Figure 7(b1,b2) show the four-layer LSTM predicted axle box bearing temperature and actual axle box bearing temperature comparison curves in the test set. Compared with the three-layer LSTM network, the error was smaller. In particular, the RMSE (root mean square error) of the predicted temperature and the actual temperature was lower than 1 °C, which is acceptable. The RMSE can be seen in Equation (10). Figure 7(c1,c2) are the five-layer LSTM comparison charts in the test set. It shows that the error was getting larger again, which means overfitting.
(10)$$\mathrm{RMSE}=\sqrt{\frac{1}{m}\sum_{i=1}^{m}{\left(y_{i}-{\hat{y}}_{i}\right)}^{2}}$$

After comparison, it can be found that there is underfitting in one, two, three-layer LSTM network, but overfitting in five-layer LSTM. The best performance is four-layer LSTM network and the fitting error is not exceeding the sampling error. Eventually, our MLSTM chooses the four-layer LSTM network structure and the detailed error of different structures can be seen in Table 2.

### 3.3. The Selection of the Unsupervised Algorithm 

The predicted temperature needs to be classified by unsupervised classification algorithm, which can eliminate the subjective effect of labelling. In this paper, four kinds of unsupervised classification algorithms were compared to obtain the best-unsupervised classification algorithm. They were the iForest algorithm, LOF (local outlier factor) algorithm, Kmeans algorithm, and one-class SVM (support vector machine) algorithm.

The iForest algorithm has three core parameters. The first one is the height limit *H*, which was set to 6. The second parameter is the number of iTrees *N*; since the iForest algorithm uses ensemble learning, the larger number has better performance, but it will affect the calculation speed. After the experiment, it was compatible with speed and accuracy when it was set to 200. The abnormal score threshold is the most core parameter of this algorithm, which was set to 0.772. 

The core parameter of a one-class SVM is the kernel function, where the linear kernel function was selected as the kernel function because the fault detection of axle box bearing belonged to a two-classification problem. K is number of neighbors in the LOF, where a false warning will occur if the value is small and will miss the warning otherwise; K was selected to be 6. K is the number of clusters in Kmeans and 2 has been selected as the K.

After inputting the degree of deviation $index_{s}$ into each unsupervised classification algorithm, the accuracy on the training set and test set was compared. In Table 3, it is shown that MLSTM-iForest had the best performance. The definition of accuracy can be seen in Equation (11), where $N_{c}$ represents the number of faulty bearing found in the inspection and $N_{w}$ represents the number of unsupervised algorithm warnings.
(11)$$\mathrm{Accuracy}=\frac{N_{c}}{N_{w}}$$

The length of a single sample in the test set was 9848. The false warning rate, miss warning rate, and time cost of each classification algorithm were compared, where the average indicators on the test set can be seen in Table 4. A false warning refers to the failure found in the inspection but the classification algorithm did not report it, while the miss warning is just the opposite. Through the comparison, the false warning rate of the MLSTM-LOF and MLSTM-One-Class SVM was too large, the MLSTM-iForest and MLSTM-Kmeans had no missing warning, but the false warning rate of MLSTM-Kmeans was more than twice that of MLSTM-iForest. Although the time cost of MLSTM-iForest was nearly ten times that of the other algorithms, the time cost was negligible compared with the early warning time. Through the comparison, the iForest algorithm was selected as the unsupervised classification algorithm. 

### 3.4. MLSTM-iForest Algorithm

In order to realize the early warning for an axle box bearing fault, the MLSTM-iForest algorithm was proposed, which can be seen in Algorithm 1. The inputs of the presented method are train speed marked as $S_{t,i}$, motor traction power marked as $P_{t,i}$, carriage mass marked as $M_{t,i}$, ambient temperature marked as $O_{t,i}$, axle box bearing temperature marked as $T_{t,i}$, iForest height limit marked as *H*, the number of trees in iForest marked as *N*, and the abnormal score threshold marked as *A*. The output of MLSTM-iForest is the axle box bearing classification result marked as $R_{t,i}$, which only has two possible results, normal or fault. In the algorithm, subscript *t* means “at time *t*,” *i* is the carriage number, superscript *n* stands for normalization, *p* means prediction, and *s* is the sliding window step. The detailed steps of the algorithm are described as follows.

Step 1: The first step of the algorithm is preprocessing, which is done to standardize different magnitude data into the same magnitude to improve the convergence speed and model accuracy. The *Z*-score standardized method was adopted to resolve the possibility of outlier data beyond the value range, and is shown in Equation (12), in which *x* is the data to be normalized, *μ* is the mean of the input data, and *σ* is the standard deviation of the input data. After preprocessing, all inputs are marked as $X_{t,i}^{n}$, which contains the train speed $S_{t,i}$, traction power $P_{t,i}$, carriage mass $M_{t,i}$, ambient temperature $O_{t,i}$ and axle box bearing temperature $T_{t,i}$.

(12)$$Z_{s}=\frac{\left(x-\mu \right)}{\sigma }$$

Step 2: The next step is to predict the axle box bearing temperature. The standardized time-series data $X_{t,i}^{n}$ are input into the MLSTM to get the predicted temperature $T_{t+n,i}^{p}$; the MLSTM structure can be seen in Figure 8.
**Algorithm 1**: MLSTM-iForest1:  **Inputs:**
$X_{t,i}$—time-series source data, *S*—sliding window step, *H*—height limit,2:    *N*—number of trees, *A*—abnormal score threshold3:  **Outputs:**
$R_{t,i}$ classification result4:  Standardizing $X_{t,i}$ to get $X_{t,i}^{n}$5:   Inputting $X_{t,i}^{n}$ into MLSTM to get predicted data $T_{i+n,i}^{p}$6:  Calculating the deviation index $index_{s}$ based on *S*7:  Building iForest based on $index_{s}$, *H*, and *N*8:  Outputting abnormal $Score_{t,i}$ from iForest9:  **if**
$Score_{t,i}$> A10:      $R_{t,i}$ = *abnormal*11:  **else**12:    $R_{t,i}$ = *normal*13: **return**
$R_{t,i}$
$R_{t,i}$


Step 3: This step implements the early warning. First, the deviation index is calculated. The deviation index is defined using Equation (13). In this equation, $x_{t}^{j}$ represents the predicted temperature of the MLSTM at time *t*, superscript *j* represents the index number arranged from small to large of the predicted temperature of different bearings at the same location and the same time *t*. The smaller index number gives the lower temperature, that is, $x_{t}^{j-1}<x_{t}^{j}$. *T* means “at time *T*” and *s* is a sliding window step. The purpose of adding a sliding window step was to reduce the influence of abnormal data caused by the sensor suffering from a sudden disturbance of common occurrence in operation, which may bring about a misclassification. The selection of *s* was too small to eliminate the sudden disturbance of sensor and too large to lead to an error classification, where 5 was considered an appropriate choice. Second, $index_{s}$, height limit *H*, and the number of trees *N* is used to build iForest. Finally, the $Score_{t,i}$ from iForest is output; if $Score_{t,i}$ is larger than the abnormal score threshold *A,* it means the axle box bearing is abnormal, otherwise it is normal. The flow chart of this algorithm can be seen in Figure 9. In the flow chart, $S_{t,i}$ stands for train speed, $P_{t,i}$ stands for traction power, $M_{t,i}$ stands for carriage mass, $O_{t,i}$ stands for ambient temperature, and $T_{t,i}$ stands for axle box bearing temperature. The superscripts *n* and *p* represent normalization and prediction, respectively.
(13)$$index_{s}(s)=\frac{\sum_{t=T}^{T+s}\left(\frac{\sum_{j=1}^{7}x_{t}^{j}}{7}-x_{t}^{7}\right){}^{2}}{s}$$

## 4. Results and Discussions

### 4.1. Eliminating Missing Warnings

According to the statistics, the normal temperature of the axle box bearing of high-speed EMU was under 100 °C, and most of them exceeded 100 °C in the case of failure. Therefore, the threshold of an axle box bearing temperature of EMU-ODS was 105 °C, but the bearing temperature did not reach 100 °C in the case of a few failures.

The fault of a bearing cannot be judged simply by the temperature threshold because the earlier failure will not reach the threshold temperature at all. From the correlation analysis shown in Figure 5, it can be seen that the train speed, ambient temperature, and traction power had the greatest impact on the axle box bearing temperature, with the correlation coefficient exceeding 0.6. Meanwhile, the traction power had a strong correlation with the train speed, such that the train speed and the ambient temperature were the main factors affecting the axle box bearing temperature.

At the same time, the running speed and the ambient temperature of the bearings at the same position on the same side of different carriages showed little difference. If the bearings were normal, the temperature of each bearing should be close. Therefore, if the temperature of one bearing deviated too much from that of other bearings, it can be considered as abnormal and using Equation (13), the deviation index can be calculated.

From Figure 10, the curves of different colors represent the temperature curves of axle box bearings in different carriages at the same position. The red stars in the figure represent the MLSTM-iForest warning, and the red dotted line is the threshold line of 105 °C. It can be seen that the temperature deviation degree of a bearing in carriage 3 between 12:00 and 15:00 was large, but it did not exceed the threshold temperature of 105 °C and the EMU-ODS did not report a warning. The warning of the MLSTM-iForest is verified by the inspection after the operation.

### 4.2. Warning Ahead of EMU-ODS

In Figure 11, the EMU-ODS gave a warning because the axial bearing of carriage 3 exceeded 105 °C, but the warning time of MLSTM-iForest was 11 min earlier than that of EMU-ODS, which could save more time for an emergency response. According to the statistics, the MLSTM-iForest warning time should be at least 9 min earlier than EMU-ODS.

In Figure 12, the MLSTM-iForest gave a warning, while the EMU-ODS showed that the train was running in good condition on that day. However, 2 days later, EMU-ODS triggered the warning, which can be seen in Figure 13. Therefore, compared with EMU-ODS, the proposed method could give an earlier warning.

## 5. Conclusions

In this paper, a novel early warning method is presented to effectively provide early warnings of axle box bearing faults based on operation data, namely MLSTM-iForest, where MLSTM takes relevant data as input to realize the axle box bearing temperature prediction and iForest resolves the unsupervised classification, which determines whether the bearing is faulty. In order to eliminate the instantaneous abnormal data collected in the operation that causes false warnings, the degree of deviation index was proposed. The results indicated that the proposed method could effectively eliminate false warnings; furthermore, compared with other machine leaning algorithms, the accuracy of the proposed method was 98.4%, which was the best performance. Meanwhile, when the proposed method was compared with EMU-ODS, the proposed method provided warnings at least 9 min earlier and at most 2 days earlier, which could save more time for an emergency response. 

## Figures and Tables

**Figure 1 sensors-20-00823-f001:**
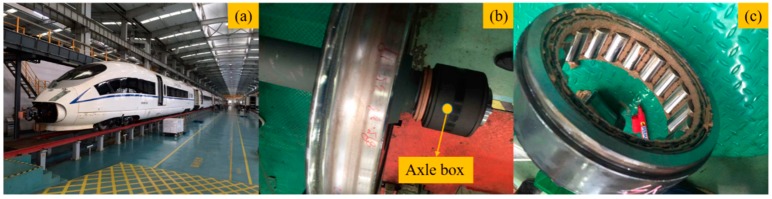
(**a**) A CRH380B high-speed EMU (electric multiple unit) under maintenance. (**b**) Installation position of axle box bearing on wheelset that was removed from the train. (**c**) A disassembled faulty axle box bearing.

**Figure 2 sensors-20-00823-f002:**
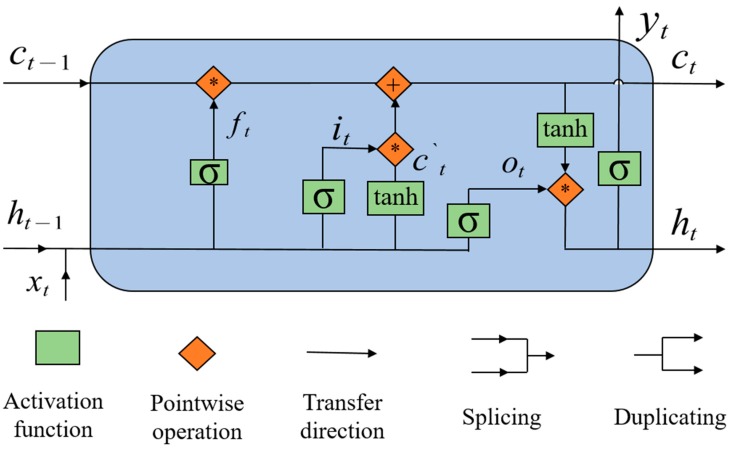
The structure of LSTM (long short-term memory) machine learning.

**Figure 3 sensors-20-00823-f003:**
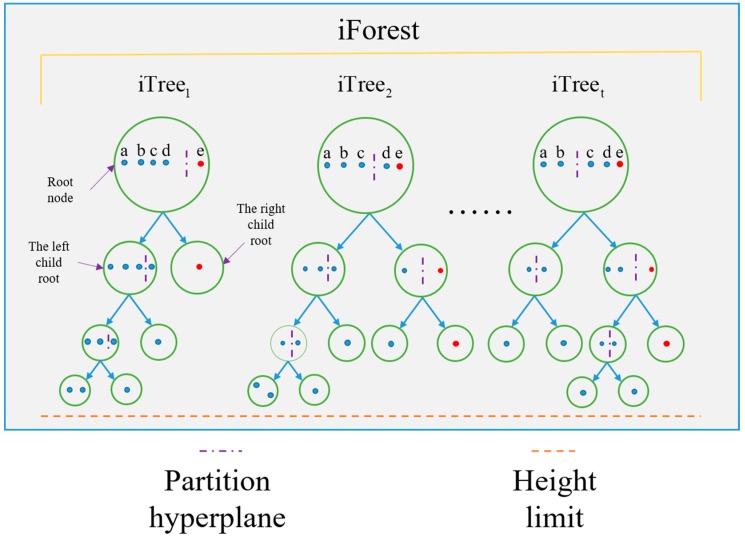
One-dimensional data used to demonstrate the principle of iForest.

**Figure 4 sensors-20-00823-f004:**
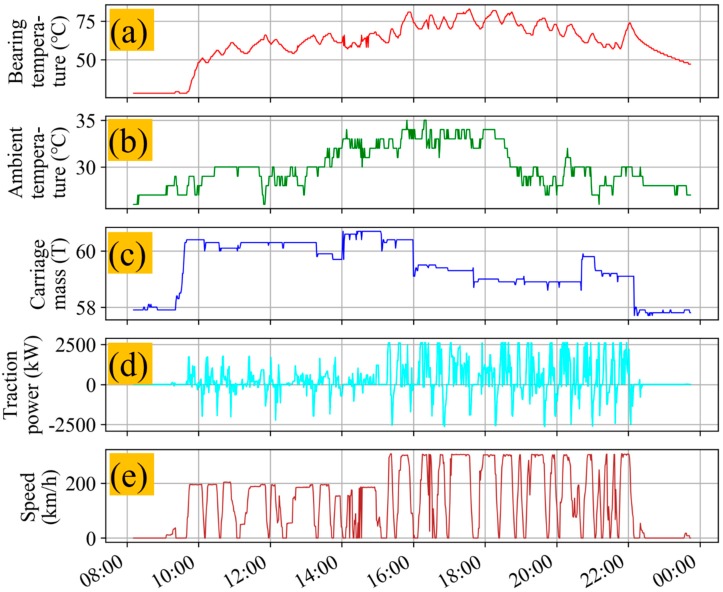
Graphs of (**a**) axle box bearing temperature, (**b**) ambient temperature, (**c**) carriage mass, (**d**) motor traction power, and (**e**) train speed collected over one day.

**Figure 5 sensors-20-00823-f005:**
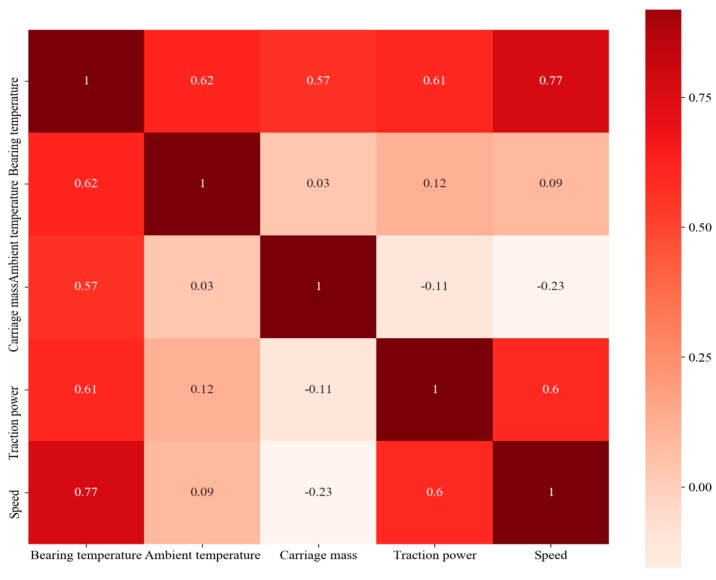
Correlation coefficients and heat map between the axle box bearing temperature, ambient temperature, carriage mass, motor traction power, and train speed.

**Figure 6 sensors-20-00823-f006:**
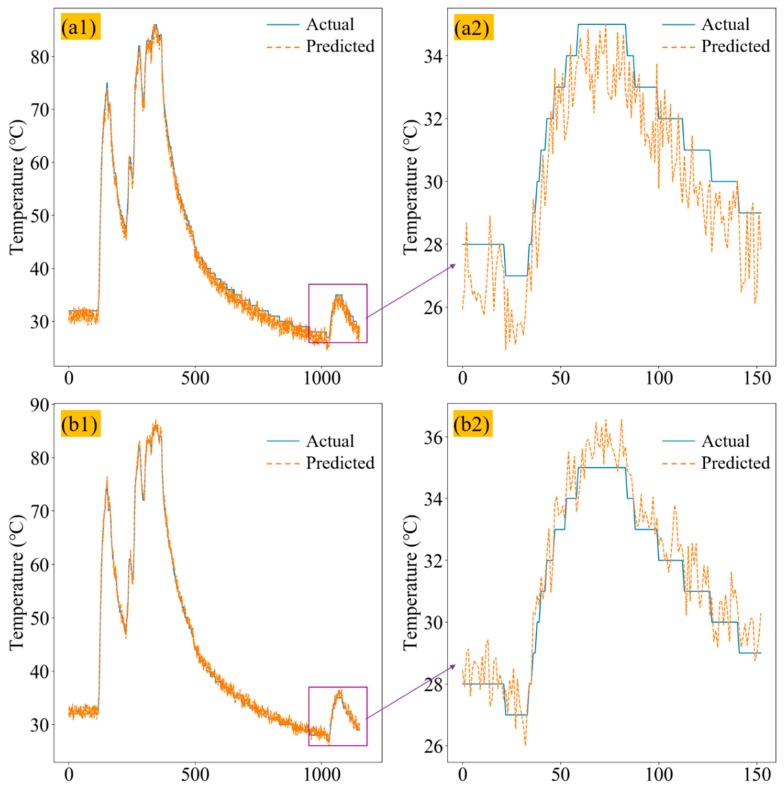
(**a1**) One-layer LSTM predicted axle box bearing temperature and actual axle box bearing temperature comparison curves. (**b1**) Two-layer LSTM actual axle box bearing temperature and actual axle box bearing temperature comparison curves. (**a2**) and (**b2**) are their local enlarged drawings.

**Figure 7 sensors-20-00823-f007:**
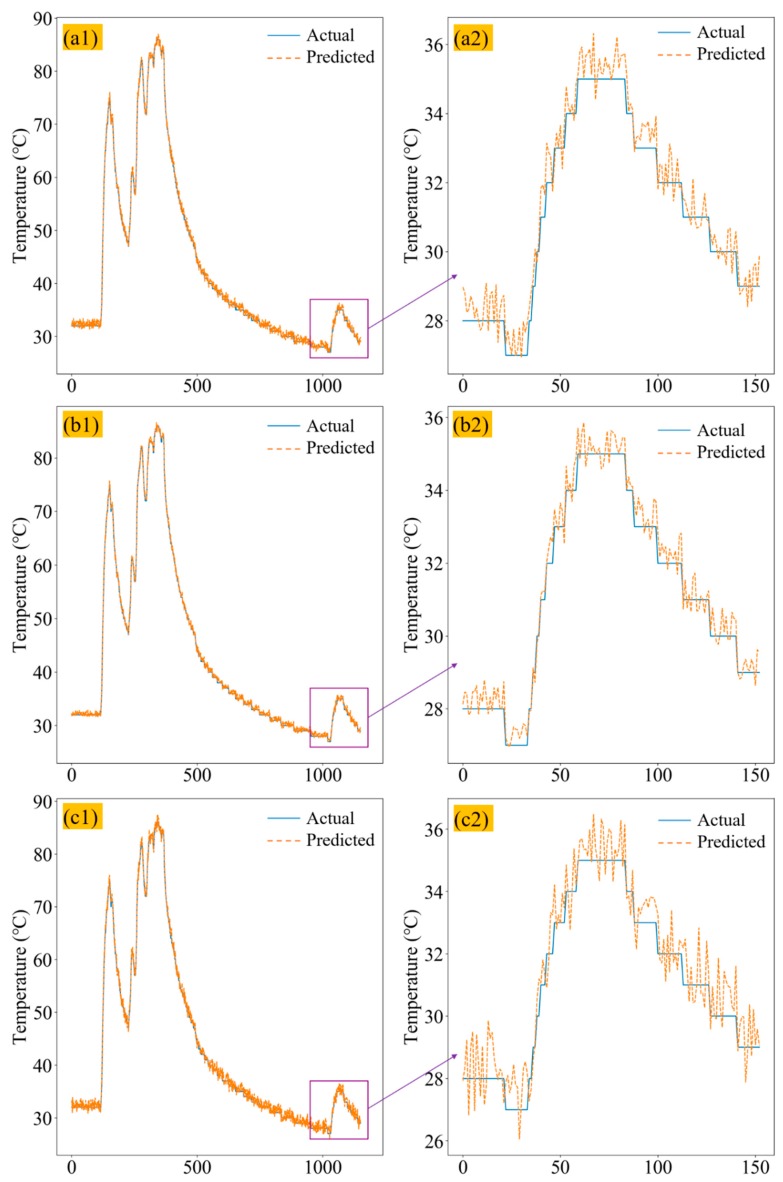
(**a1**) The three-layer LSTM predicted axle box bearing temperature and actual axle box bearing temperature comparison curve. (**b1**) The four-layer LSTM predicted axle box bearing temperature and actual axle box bearing temperature comparison curve. (**c1**) The five-layer LSTM predicted axle box bearing temperature and actual axle box bearing temperature comparison curves. (**a2**), (**b2**), and (**c2**) are their local enlarged drawings.

**Figure 8 sensors-20-00823-f008:**
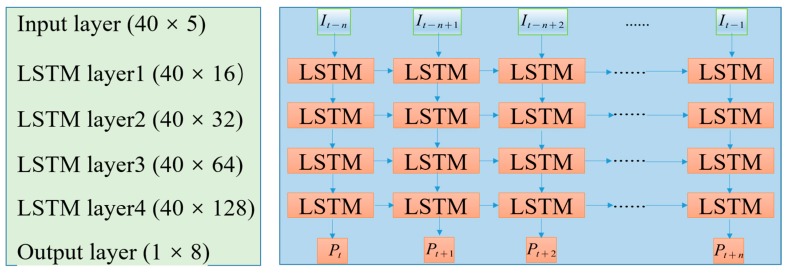
The detailed parameters of each layer of MLSTM.

**Figure 9 sensors-20-00823-f009:**
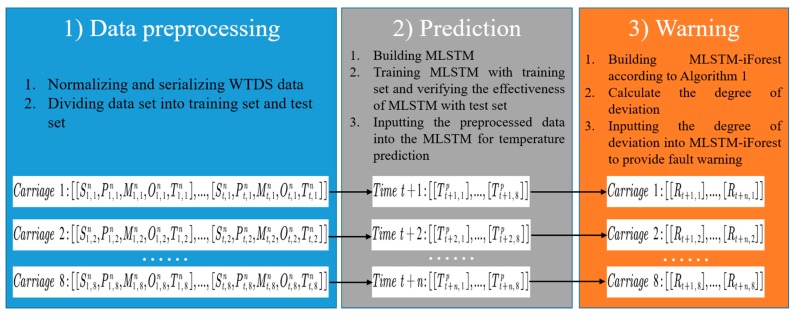
The flow chart of MLSTM-iForest. WTDS: Wireless transmit device system.

**Figure 10 sensors-20-00823-f010:**
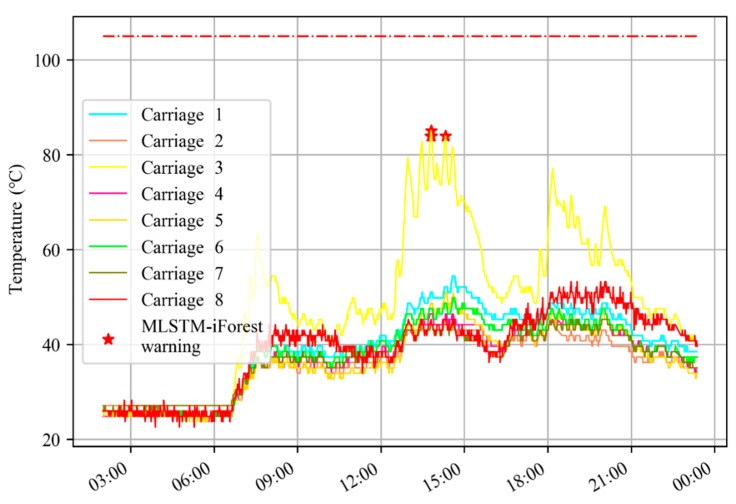
MLSTM-iForest reported warnings but the EMU-ODS (electric multiple unit onboard detection system) missed warnings.

**Figure 11 sensors-20-00823-f011:**
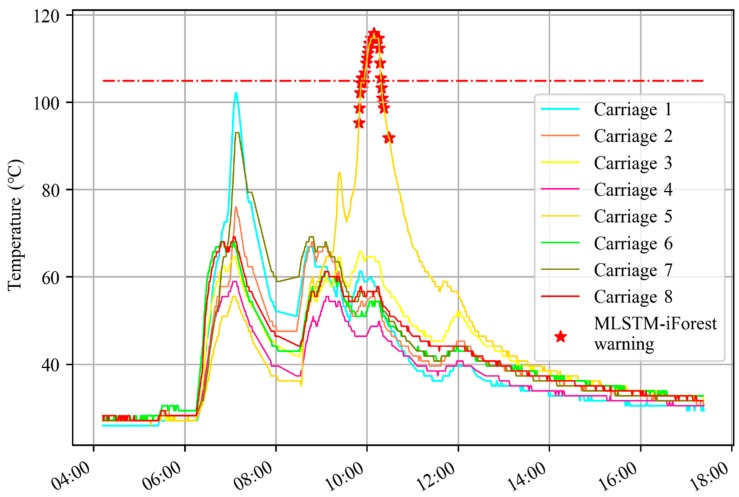
MLSTM-iForest warned a few minutes ahead of EMU-ODS.

**Figure 12 sensors-20-00823-f012:**
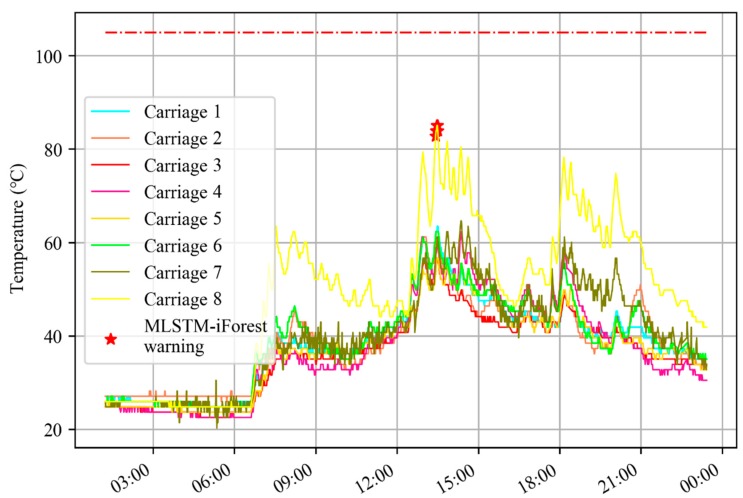
MLSTM-iForest warned two days ahead of EMU-ODS.

**Figure 13 sensors-20-00823-f013:**
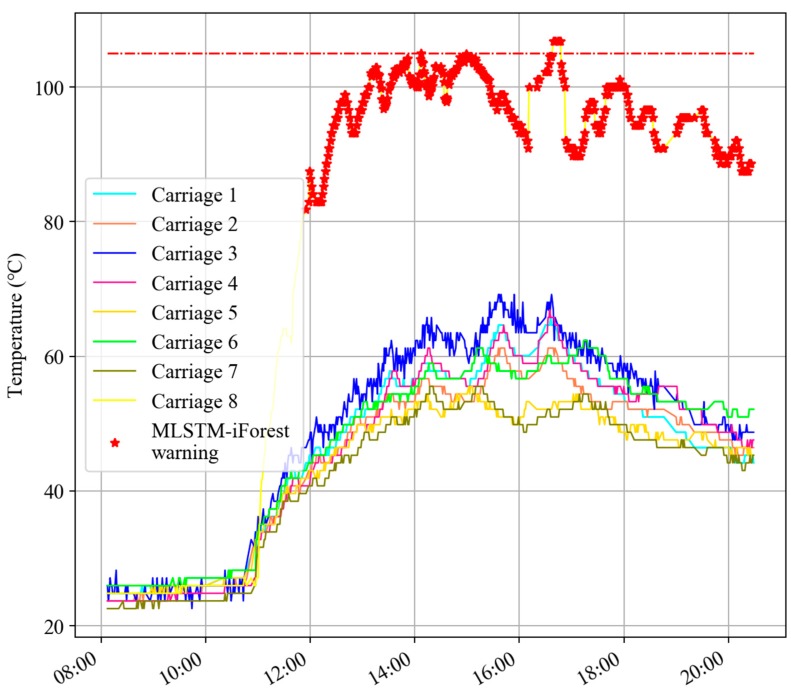
MLSTM-iForest and EMU-ODS warnings over two days.

**Table 1 sensors-20-00823-t001:** Experiment samples.

Bearing Type	Training Set	Test Set	Sample Length
Normal	60	20	480
Abnormal	60	20	480

**Table 2 sensors-20-00823-t002:** The RMSE of different layers of the LSTM for the training set and test set.

RMSE	One-Layer	Two-Layer	Three-Layer	Four-Layer	Five-Layer
RMSE for training set	1.35	0.81	0.59	**0.40**	0.78
RMSE for test set	2.41	1.64	1.21	**0.83**	1.57

**Table 3 sensors-20-00823-t003:** Accuracy comparison of the unsupervised algorithm. MLSTM: Multilayer LSTM, LOF: local outlier factor, SVM: support vector machine.

Data Set	MLSTM-iForest	MLSTM-LOF	MLSTM-Kmeans	MLSTM-One-Class SVM
Training set	98.9%	87.3%	97.8%	88.8%
Test set	98.4%	85.4%	96.2%	81.4%

**Table 4 sensors-20-00823-t004:** Indicator comparison for unsupervised algorithms.

Indicators	MLSTM-iForest	MLSTM-LOF	MLSTM-Kmeans	MLSTM-One-Class SVM
False warning rate	1.6%	13.1%	3.8%	13.4%
Missing warning rate	0%	1.5%	0%	5.2%
Time cost	1.56 s	0.17 s	0.16 s	0.14 s

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
