# Peer review of "A Real-Time Fault Early Warning Method for a High-Speed EMU Axle Box Bearing"

_sensors, 2020, doi:10.3390/s20030823_

Round 1

Reviewer 1 Report

This article presents a real-time fault early warning method for high-speed EMU axle box bearing. In order to solve the current problems of the fault early warning of axle box bearing, the authors proposed a method without installing vibration sensor on the high-speed EMU in service, namely MLSTM-iForest. It is very interesting. However, it has some problems that need to be solved further. 1. Many abbreviations are used but not defined in the abstract, such as EMU, TADS, EMU-ODS and MLSTM. 2. Some mathematical symbols are wrong in Figure 2, such as tanh? Similar problems arise in a large number of formulas. 3. Some Figures are not clear, so it is difficult for reviewers to judge the effectiveness of the method, such as Figure 7 and 8. 4. It is also difficult to prove the effectiveness of the proposed method due to the lack of comparison of similar technologies. 5. The authors are suggested to quote the latest references and standardize the format of the article. 6. In addition, in order to highlight the innovation of the paper, the advantages of the proposed method and the numerical results of the experiment should be added in the conclusion. In view of the above problems, this article cannot be published in Sensors in its current form.

Author Response

    Firstly, thank you so much for your reviews. Then, I will answer your comments one by one. 1.The acronym problems have been revised. 2. I can't understand why the mathematical symbol tanh is wrong and tanh can be seen in many papers. 3. Unclear figures have been replaced by clear pictures. 4. Some machine learning algorithms have been added for comparision. 5. The latest references have been added and standardized the reference format. 6. The conclusion have been modified. 

Reviewer 2 Report

In this paper, the authors proposed a real-time fault early warning method for high-speed EMU axle box bearing. According to statistics, the number of high-speed EMU in China is increasing. It is very important topic to study the real-time fault early warning of high-speed EMU axle box bearing. The MLSTM-iForest algorithm was introduced to this research. The research result is valuable to the related research. The authors are encouraged to make following revision.

(1) In order to express the algorithm of the paper more clearly, a flow chart should be added to the paper.

(2) In Figure 11, the algorithm shows a warning before 2 days. Please also give the figures in next 2 days if possible.

(3) The conclusion of this paper should be more in-depth description, not only the conclusion of the experiment.

Author Response

     Thank you very much for your reviews. I'm so sorry for the late reply.  I will answer your comments one by one. 1. The flow chart have been added. 2. the comparison picture of 2 days later have been added. 3. The conslusion have been modified.

Reviewer 3 Report

This manuscript proposes a fault detection methodology for High-Speed Electric Multiple Units Axle Box Bearing. The proposal is interesting and the experimental evaluation indicates the advantages of the proposed methodology. However, there are some issues that must be considered for improving this paper.

1) English must be revised.

2) In my opinion, it is lacking further discussions on the related state-of-the-art fault detection techniques. There is a discussion on other techniques and other instrumentation strategies for fault detection, but in my opinion, the most recent and efficient ML-based fault detection techniques are not discussed.

3) Is the experimental comparison with the so-called EMU-ODS fair? Why not comparing with other recently published ML-based fault detection techniques?

4) The contribution list should be improved. The list is repetitive and does not make clear what are the technical contributions of this paper over the literature. It is important to notice that the contribution statement is not a summary of the presented work.

5) Is the MLSTM-iForest an original contribution of this paper? Or is the application of this algorithm for fault detection the original contribution? In the first case, please, clarify the differences between the proposed algorithm and the other MLSTM in the literature? Is it simply the use of the common MLSTM with isolation forest for fault detection?

6) In my opinion, the explanation about MLSTM-iForest is unclear and confusing. If it is one of the main contributions, it deserves more attention. However, the explanation is mixed with experimental data that distracts the reader. I think the section must be revised in order to focus on the methodology explanation. There are elements for it, since Algorithm 1 and the steps of the methodology are presented in the current version. In addition, I suggest incorporating the steps 1-3 into algorithm 1, aiming to make it self-contained.

7) How is the Abnormal score threshold (A) computed?

8) Minor issues

- There is a lot of acronyms and some of them are undefined. I suggest to drag the list of acronyms for the beginning of the text. Furthermore, every acronym must be defined in its first use.

- p.5, l.179: there is a reference error.

Author Response

Thank you very much for your reviews. I'm so sorry for the late reply.  I will answer your comments one by one. 1. English expression have been modified. 2. the efficient machine learning algorithm have been added as comparision. 3. Currently, only EMU-ODS are used on high-speed EMU for condition monitoring.4. The contribution have been improved. 5-6. MLSTM-iForest is an original algorithm and the algorithm description have been revised. 7. The abnormal score threashold from statistics of the experiment results. 8. Minor problems have been revised.

Reviewer 4 Report

The paper presents an alternative approach to high-speed EMU axle box bearing. The paper starts with the problem presentation and a very timid overview of the related literature. After that, the proposed model is presented; and the proposed approach is described. And then, the results of some experimental tests are carried out without comparisons with other approaches. The paper has good contributions; however, the authors are asked to some topics in their paper:

1) the literature overview must be more complete and it must contain the major drawbacks; and

2) the results must contain comparisons with other similar approaches existing in the literature.

Author Response

Thank you very much for your reviews. I'm so sorry for my late reply. I will answer the comments one by one. 1.The literature overview have been modified. 2. Some machine learning algorithms have been added as comparison.

Reviewer 5 Report

The paper deals with a “method to realize the fault early warning of axle box bearing”. In general, the manuscript is not well written: 1) the state of the art has not been enough explored; 2) the paper is written in first and third person while an academic manuscript generally should be written in third person; 3) in the text, there are instances of usage of inappropriate words, unclear statements, grammatical errors and variable tenses (row n. row n. 19 “time series …is..”; row n.21 past tense after that you used present tense); 4) the quality of figures is poor (see Fig. 4) and the graphs are not showed properly: the scales for the axes are very small (see Fig. 7 and Fig. 8); 5) images are used instead of a font (see rows n. 250, 253, 254, 258). Therefore, I invite the authors to fix the several highlighted issues and revise the paper. For above said, I recommend a major review before the publication in Sensors.

Author Response

Thank you very much for your reviews. I'm so sorry for my late reply. I will answer the comments one by one. 1. the state of the art have been modified. 2. the  paper is written in third person. 3. the mentioned problems have been revised. 4. The quality of figures have been improved and it's still a whole day's data in Fig.7 and Fig.8, but the abscissa is time-series of data rather than time. 5. I can't understand this review.

Round 2

Reviewer 1 Report

The authors have adequately addressed part of my concerns in the review, and did a good job to revise and improve the paper. However, there are still some problems to be solved.

Line 1, Type of the Paper (Article, Review, Communication, etc.). The authors needs to choose one. Line 2, the font size is incorrect. The format of many formulas is wrong. For example, the W and b in formula 2 and 3 are different, which one is right? The symbol * is not defined.
All the Formula mentioned in this paper should use "Formula 1", "Formula 2", and so on. All Figures should be centered, and the format of all the Tables are not standard, which should be fixed.
Section 3.3 has many problems in typesetting. Figure 9 is still not clear. More analysis and discussion should be added in Section 4.1.
Figure 61, 72, 83? The format of references is still inaccurate, such as [21], [25] and [26].

In a word, the authors didn't revise this paper carefully, it cannot be published in Sensors in its current form.

Author Response

Dear professor,

     Thanks so much for your patience to reply. The type of this paper is Article which have been chosen from my first submit and I don't know why the download manuscript does't show the paper type; I can't control the font size in Line 2, I think this is editor's work. The format of Formula, Figure and Table have been fixed. The definition of "*” was given and can be seen above Figure 2. The typesetting problems of 3.3 have been modified. Figure 9 have been changed. More analysis and discussion have been added in Section 4.1.The format of references in [21], [25] and [26], I can give a subjective abbreviation of the journal which are maily conference paper, and this may be the work of editor.

Reviewer 3 Report

In my opinion, the current version is improved. However, the issues that I've indicated were not adequately answered. It is not clear why it is no possible to compare the results with other ML-based approaches and the originality of contribution is still unclear.

Author Response

Dear professor,

    Thanks for our patience to reply. The comparison with other ML-based method can be seen in section 3.3. The contribution have been modified.

Reviewer 4 Report

No further comments.

Author Response

Dear professor,

      Thank you so much for your patience.

Reviewer 5 Report

The authors implemented many corrections based on the previous review. However, the quality of Figure 9 is very low. In Figure 6 and Figure 7 the legend is overlapped to graph. I suggest a minor revision to fix these issues.

Author Response

Dear professor,

    Thanks so much for your patience to reply.The mentioned problems have been modified.